# Does the neuronal noise in cortex help generalization?

**Brian Hu**
Allen Institute for Brain Science
`brianh@alleninstitute.org`

**Jiaqi Shang**
Allen Institute for Brain Science
`jiaqi.shang@alleninstitute.org`

**Ramakrishnan Iyer**
Allen Institute for Brain Science
`rami@alleninstitute.org`

**Josh Siegle**
Allen Institute for Brain Science
`joshs@alleninstitute.org`

**Stefan Mihalas**
Allen Institute for Brain Science
`stefanm@alleninstitute.org`

## Abstract

Neural activity is highly variable in response to repeated stimuli. We used an open dataset, the Allen Brain Observatory, to quantify the distribution of responses to repeated natural movie presentations. A large fraction of responses are best fit by log-normal distributions or Gaussian mixtures with two components. These distributions are similar to those from units in deep neural networks with dropout. Using a separate set of electrophysiological recordings, we constructed a population coupling model as a control for state-dependent activity fluctuations and found that the model residuals also show non-Gaussian distributions. We then analyzed responses across trials from multiple sections of different movie clips and observed that the noise in cortex aligns better with in-clip versus out-of-clip stimulus variations. We argue that noise is useful for generalization when it moves along representations of different exemplars in-class, which is similar to the structure of cortical noise.

## 1 Introduction

One of the hallmarks of neural codes is the high level of trial-to-trial variability [1, 2]. This variability has been studied using multiple stimuli [3], along with its relation to attention [4] and other behavioral variables [5]. Previous theories on the possible role of noise center on its potential usefulness in inference [6]. In the field of machine learning, noise can have a regularizing effect and enable better model generalization (e.g. dropout [7]). Here, we explore the hypothesis that networks of cortical neurons use noise with the goal of building general representations from a small number of exemplars.

First, we show that cortical noise is often non-Gaussian, and better captured by long-tailed distributions or mixtures of Gaussians. This result was consistent across experiments using two-photon calcium imaging and electrophysiological recordings. To control for possible state-dependent effects, we used a population coupling model where the activity of all other simultaneously recorded neurons is used to predict the activity of a single neuron (and as a proxy for brain state). Finally, we defined a set of neural subspace measures and found that cortical noise aligns with in-class stimulus variations.

Real Neurons and Hidden Units Workshop at the 33rd Conference on Neural Information Processing Systems (NeurIPS 2019), Vancouver, Canada.

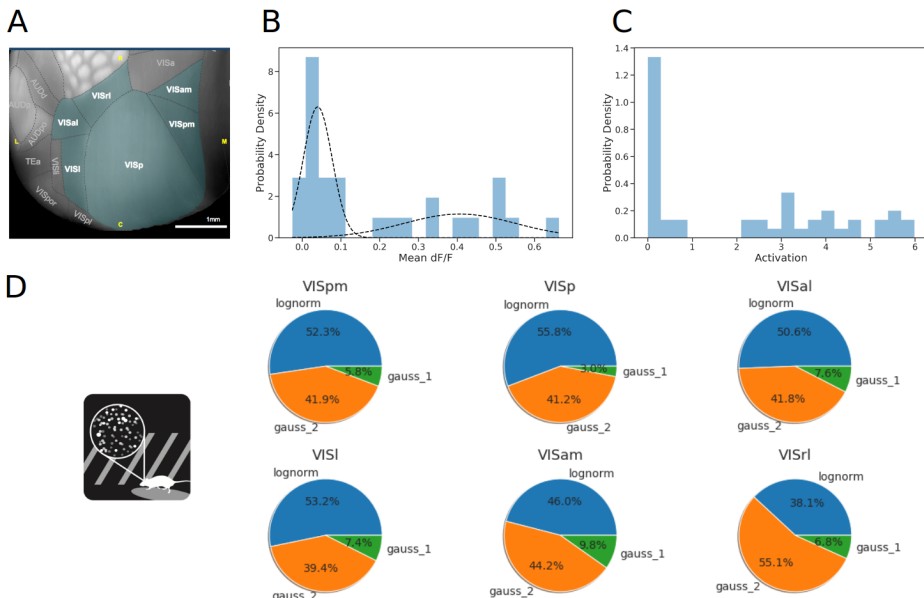

Figure 1: Neural variability in the Allen Brain Observatory. (A) Visual areas where recordings were performed. (B) The distribution of responses for an example cell. The two components of the Gaussian mixture are shown in dotted black lines. (C) The distribution of activations for an example unit in the neural network (second convolutional layer, fourth feature map). (D) Percent of cells in each of the recorded visual areas, broken down by the best-fitting distribution.

## 2 Results

**Noise distribution in the Allen Brain Observatory [8].**    We used the 30 s long "natural movie one" stimulus. The movie was presented 10 times over 3 imaging sessions ($N = 30$ trials). We analyzed all excitatory cells in the dataset, excluding cells with a mean trial-to-trial correlation below zero, which resulted in $N = 11,428$ cells. As an additional control to remove extremely unreliable cells, we only analyzed cells above a minimum trial-to-trial correlation threshold of 0.1, which yielded qualitatively similar results (see Appendix for details). We split the movie into non-overlapping 1 s epochs, and computed the mean change in calcium fluorescence response relative to baseline ($dF/F$) for each cell over each epoch. The "preferred" stimulus for each cell was defined as the epoch which elicited the max mean $dF/F$ response over all trials.

We analyzed trial-to-trial variability in neural responses to each cell's "preferred" stimulus across visual areas. We find that the majority of cells are best fit by log-normal distributions or Gaussian mixtures with two components, with 40.4% of cells showing dropout-like distributions (see Methods and Appendix for more details). We performed a similar analysis on units from a convolutional neural network[1] trained on CIFAR-10 [9], using each unit's "preferred" image. Dropout ($p = 0.5$) was used in all layers during training and inference, which may act as a form of Bayesian approximation [10]. We find that two-component Gaussian mixtures can also capture the responses from the network with dropout (see Appendix for more details). Figure 1 shows example response distributions, and a summary of the distribution fits across cells in our dataset.

**Control for eye movements.**    A potential source of variability is eye movements. In primates, small fixational eye movements can drive trial-to-trial variability and correlations between neurons [11]. Mice, which do not have a fovea, have much lower visual acuity and may move their heads more than their eyes in order to view different parts of the visual scene [12]. Furthermore, bilateral eye

---

[1]Network architecture: conv5-10, conv5-10, conv5-20, conv5-20, fc-320, fc-50, where 'conv' represents convolutional layers (<kernel size>-<number of features>) and 'fc' represents fully connected layers (<number of features>). Max pooling was used after every second convolutional layer. The model achieved a test accuracy of $\sim 60\%$, which we did not optimize for as this was not the focus of our paper.

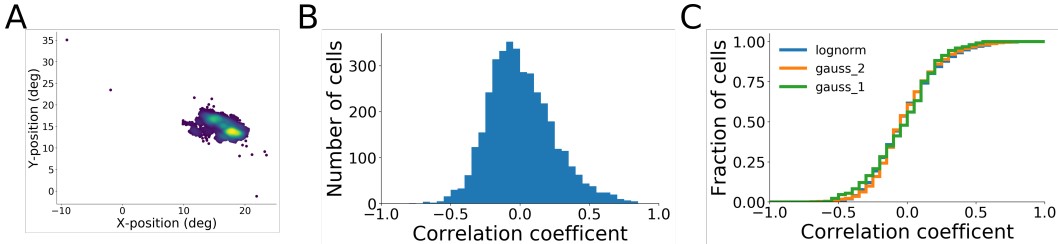

Figure 2: Quantification of eye movements during repeated natural movie presentations. (A) Eye positions recorded for an example mouse during repeated natural movie presentations from the Allen Brain Observatory. The mouse's gaze position was relatively localized on screen coordinates. (B) Distribution of correlation coefficients between eye position and neural activity for all cells with associated eye tracking information, which shows a mean close to zero. (C) Cumulative distribution of correlation coefficients for the cells in panel (B), conditioned on their best-fitting distribution (lognormal, two-component Gaussian mixture, or one-component Gaussian). There were no significant differences in the cumulative distributions of correlation coefficients.

movements have been found to be more active and divergent in freely moving compared to head-fixed mice [13]. In all our experiments, mice were head-fixed, and we found that eye positions were relatively stable over the course of the experiment (mean standard deviation = 3.03 deg, which is small compared to the median V1 receptive field size of ∼10 deg [14], Figure 2A). We quantified the trial-to-trial correlation between eye position and neural activity, and found that the mean correlation coefficient across all cells was not significantly different from zero ($p < 0.05$, Figure 2B). If eye movements are the source of differences in the observed response distributions, we should see differences in the correlations between eye position and neural responses based on each cell's best-fitting distribution. However, we did not find significant differences between the correlations for cells conditioned on their best-fitting distribution (KS test, $p > 0.05$, Figure 2C). An alternative approach to control for eye movements is to use the gaze position as an additional signal when fitting predictive models of neural responses, for example by using a shifter network [15], however given the very small variations compared to receptive field sizes present, we did not pursue this.

**Noise distribution and state dependence in electrophysiological recordings.** In vivo recordings were performed in the visual cortex of awake, head-fixed mice using Neuropixels probes [16]. The repeated natural movie stimulus was 81 s long, consisting of 11 shorter clips ranging from 4 to 9 s each ($N = 98$ trials). All spike data were acquired with a 30-kHz sampling rate and recorded with the Open Ephys GUI. A 300-Hz analog high-pass filter was present in the Neuropixels probe, and a digital 300-Hz high-pass filter (3rd-order Butterworth) was applied offline prior to spike sorting. Spike times and waveforms were automatically extracted from the raw data using Kilosort2. After filtering out units with "noise" waveforms using a random forest classifier trained on manually annotated data, all remaining units were packaged into the Neurodata Without Borders format for further analysis. This resulted in a total of $N = 936$ units across three mice.

One potential source of variability is state-dependent changes in neural activity [5], which we control for by using a population coupling model (see Methods). We again analyzed trial-to-trial variability by fitting different distributions to the neural response residuals from this model. We find that the majority of cells are still better fit by either log-normal distributions or Gaussian mixtures, even when including Poisson and negative binomial distributions (Figure 3).

**Trial-to-trial variability mimics in-class exemplar changes.** We randomly choose 10 non-overlapping 200 ms long sections ("exemplars") from each of the 11 movie clips. We exclude the first 1 s following a clip transition to avoid onset transient effects. We define the activity for neuron $i$ of exemplar $j$ in clip $k$ during trial $n$ as the spike count during the presentation of the exemplar $a_{i,j,k,n}$. The signal for neuron $i$ of exemplar $j$ in clip $k$ is the average over trials $s_{i,j,k} = \langle a_{i,j,k,n} \rangle_n$. The noise for neuron $i$ of an exemplar $j$ in clip $k$ during trial $n$ is the activity minus the signal $n_{i,j,k,n} = a_{i,j,k,n} - s_{i,j,k}$. We also define the following neural subspaces:

The **exemplar coding subspace** for an exemplar and clip is defined as the set of neurons for which the signal is larger than the mean, $E_{j,k} = \{i | s_{i,j,k} > \langle s_{i,j,k} \rangle_{j,k}\}$. The **clip coding subspace**

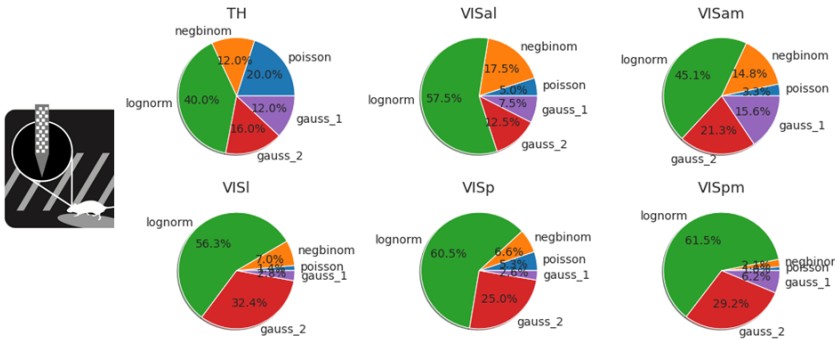

Figure 3: Neural variability in the electrophysiological recordings after controlling for state dependence. A large fraction of cells are still best fit by log-normal distributions or Gaussian mixtures.

is defined as the set of neurons for which the signal is larger than the mean for more than half the exemplars in the clip, $C_k = \{i|mean_j(s_{i,j,k} > \langle s_{i,j,k}\rangle_{j,k}) >= 0.5\}$. The **clip variance subspace** is defined as the set of neurons for which the variance of the signal for exemplars in-clip is larger than out-of-clip, $V_k = \{i|std_j(s_{i,j,k}) > std_{j,k}(s_{i,j,k})\}$. The **noise subspace** is defined as the set of neurons for which the absolute value of the noise is larger than its standard deviation, $N_{j,k,n} = \{i|abs(n_{i,j,k,n}) > std_{j,k,n}(n_{i,j,k,n})\}$.

Intuitively, the measures defined above allow us to measure the "alignment" between different subspaces used to code for the stimulus. If two subspaces are relatively aligned, they will share a common set of neurons which are active and span both subspaces, which we can quantify with a distance measure (with smaller distances being more aligned, larger distances being less aligned). Our main hypothesis is that noise for an exemplar in clip $k_1$ should lie in the clip $k_1$ variance subspace, but not the clip $k_2$ variance subspace. This allows noise to move along representations of different exemplars in the same clip, which may be useful for generalization from a small number of exemplars.

We found that the distance between the noise subspace for exemplars in clip $k_1$ is smaller for the clip $k_1$ versus the the clip $k_2$ variance subspace, suggesting that noise aligns better with in-clip variations. We also found that noise aligns with the same clip coding subspace and the exemplar coding subspace (Figure 4). We computed these measures for each mouse and visual area with at least 20 reliable neurons (9 areas passed this threshold). The differences are in the right direction and statistically significant in each individual area analyzed ($p < 0.05$). As a control (not shown here), we randomly shuffle the exemplar identity in the original data and the same analysis showed no differences in the distances between in and out classes. This suggests that there is greater alignment between the clip variance subspace and the noise subspace for exemplars in the same clip compared to other clips.

**Relation to dropout and new avenues of machine learning research.** Dropout has been shown to be an effective regularization technique that prevents model overfitting and reduces feature co-adaptation [7]. The non-Gaussian distributions we observed in the data inspired us to use a subspace analysis. As dropout-like noise generates projections in neuronal space, eliminating some neurons altogether, it is a natural place to focus on for the analysis of how trial-to-trial variability and noise aligns with different neural subspaces. Future work will study how different forms of subspace-aligned noise may help deep neural networks generalize better from fewer examples.

## 3   Discussion

In the first part of the paper, we observed complex, non-Gaussian distributions in the responses of neurons even for their preferred stimulus. In the second part of the paper, we found that trial-to-trial noise for an exemplar in a clip aligns better with exemplar-by-exemplar variation in the same clip than for other clips. We believe that research into the structure and role of biological noise will be useful for developing new methods to train neural networks with better generalization capabilities.

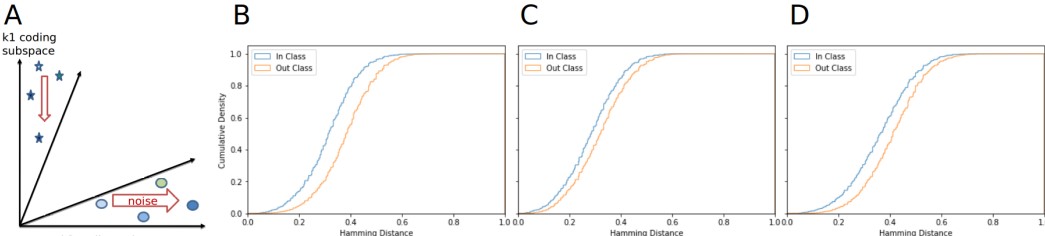

Figure 4: Noise aligns with in-clip variations. (A) Exemplars from different clips (stars and circles) activate different sets of neurons (clip coding subspace and clip variance subspace). Trial-to-trial variability due to cortical noise (red arrows) is more likely to be aligned with these subspaces (noise subspace). (B-D) Cumulative distribution of Hamming distances between the noise subspace and three other subspaces: the clip variance subspace (B), clip coding subspace (C), and exemplar coding subspace (D). Distances are lower in-class (blue) versus out-of-class (orange), suggesting a shared subspace. See definitions above for more detailed descriptions of the subspace measures.

## 4   Methods

**Noise distribution fitting.** For each cell, we quantified the distribution of neural responses across trials by fitting different distributions using the *scikit-learn* package. We fit one- and two-component Gaussian mixtures, log-normal, Poisson, and negative binomial distributions. We only fit the Poisson and negative binomial distributions to the electrophysiological recording data since these require discrete count data. We used the Akaike information criterion (AIC) to select between model fits, although other information theoretic measures yielded qualitatively similar results. We used a bootstrap parametric cross-fitting test with $N = 10,000$ samples to determine the significance of the two-component Gaussian mixture model fits. For cells best fit by the two-component Gaussian mixture, we performed an additional test to determine whether their response distributions were dropout-like. For each of these cells, we calculated a z-score on the component with the lower mean, and those cells with z-scores less than two (meaning their means are not significantly different than zero) were counted as cells with dropout-like response distributions.

**Population coupling model.** If neural variability is exclusively the result of state fluctuations, it should be captured by the coupling of each cell's activity to a lower-dimensional representation of population activity. We isolated each neuron and clustered the activity of the other neurons into 100 clusters. We used agglomerative clustering from the *scikit-learn* package with average linkage and the pairwise Pearson correlation coefficient of single-trial activities. The average activity of neurons within each cluster was used as predictors for the single-trial activity of the held-out neuron. For each neuron, we then fit a generalized linear model with the Gaussian family and an identity link function using the *statsmodels* package. We split the single-trial activities into two equal halves, using the first half for training and the second half for testing. The difference between the model predicted responses and the experimentally observed responses was used to calculate the residual activity for each neuron. The distribution of residuals was then fit using the methods outlined above.

## Acknowledgements

We wish to thank the Allen Institute founder, Paul G. Allen, for his vision, encouragement, and support.

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

## Appendix

**Additional Analysis of Allen Brain Observatory Data**

**Bootstrap parametric cross-fitting test.** As an alternative to the AIC method for testing best fit distributions, we supplement the results with a bootstrap test. For each cell, we identified its preferred "stimulus" within the movie clip and quantified trial-to-trial variability by fitting Gaussian mixture models to the distribution of responses across trials ($N = 30$) for the cell's preferred "stimulus".

For each cell, we quantified the distribution of neural responses across trials by fitting Gaussian mixture models with either one or two components to the distributions using the *scikit-learn* package. We used a bootstrap parametric cross-fitting test with $N = 10,000$ samples to determine the significance of the two component model fits. This test effectively compares how likely the difference in log-likelihoods between the one and two component models can be achieved purely by chance, given the null hypothesis that the data is generated from a single component distribution.

We find that 88.7% of the cells (10403/11731 cells) are better fit by Gaussian mixture models with two components compared to one component, indicating a two component Gaussian Mixture Model better fit the distribution in their responses compared to one component ($p < 0.05$, bootstrap parametric cross-fitting test). Model selection based on information theoretic measures such as AIC resulted in a similar proportion of cells with two component response distributions (92.0%, 10794/11731 cells). Using only a subset of the cells with reliabilities above a minimum reliability threshold (see below for details) resulted in a lower, but still substantial proportion of cells (80.6%, 1430/1775 cells). Figure 5 shows the variability in neural responses for an example cell in our dataset.

**Comparison with convolutional neural networks.** We also performed a similar analysis using the units within a convolutional neural network. For this analysis, we trained a simple network with dropout on the CIFAR-10 image dataset [9]. As this model was trained with static images rather than movies, we instead performed our analyses using the set of 118 natural images in the Allen Brain Observatory. We passed each natural image through the model 50 times, and recorded the distribution of activations for each unit within the model to its "preferred" stimulus. Normally, dropout is only

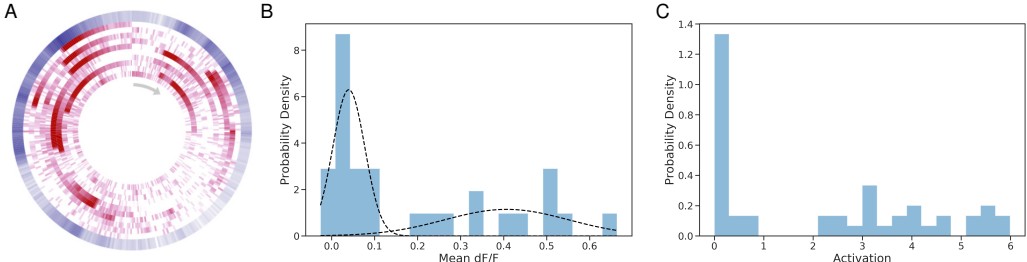

Figure 5: Neural variability for an example cell from the Allen Brain Observatory and an example unit within a neural network trained on the CIFAR-10 image dataset. (A) Track plot visualization of neural responses to the natural movie stimulus in the Allen Brain Observatory for an example cell (session A, cell specimen id 517447794). Frames of the movie are shown clockwise starting with the gray arrow, with the ten repeats within a session shown in red extending radially. The mean response across trials is shown by the outer blue ring. For many frames of the movie, the cell does not respond on every trial, even though the stimulus shown is exactly the same. (B) The distribution of responses for the example cell in (A) across all trials (three sessions with ten repeats each). The two components of the Gaussian mixture model are shown overlaid in the dotted black lines. (C) The distribution of activations for an example unit within a neural network (second convolutional layer, fourth feature map) across all trials (50 repeat presentations of natural images).

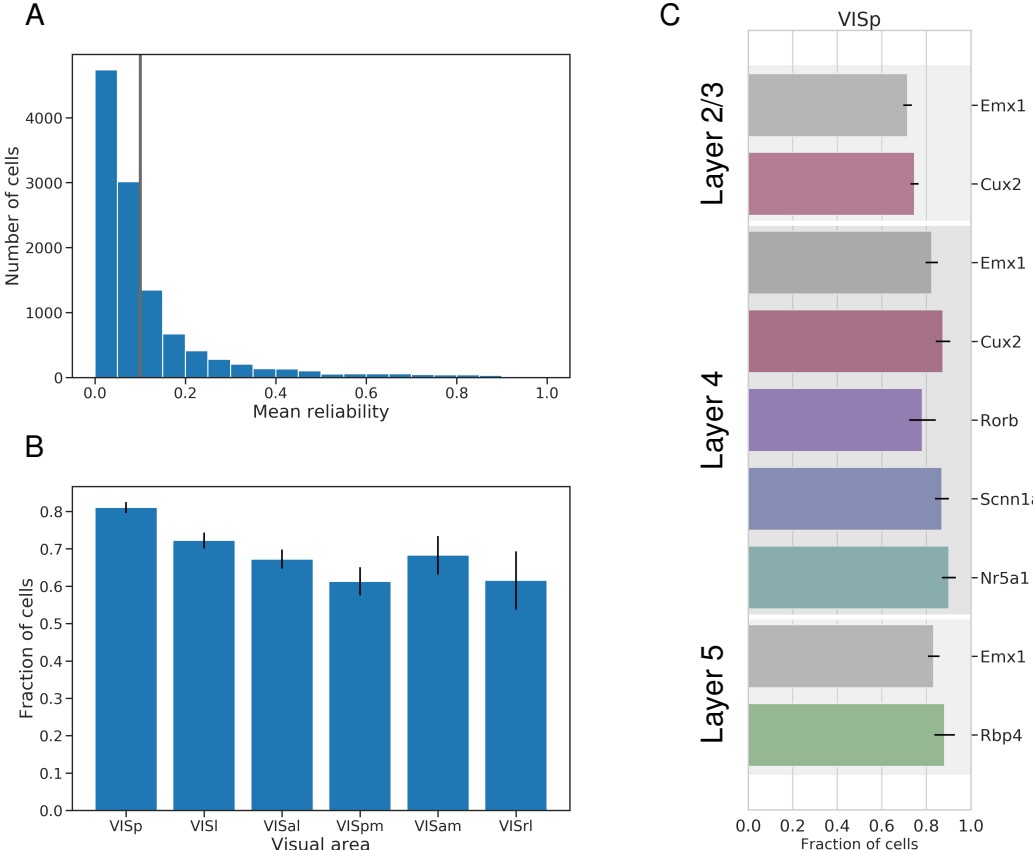

Figure 6: Quantification of neural variability in the full population of cells across visual areas, layers, and transgenic mouse lines. (A) The distribution of mean reliabilities (average trial-to-trial correlations) across the three imaging sessions for all cells included in our analysis. The vertical gray line shows the reliability threshold criterion of 0.1 which we used. (B) The fraction of cells with response distributions consistent with dropout across visual areas (mean = 0.74). Error bars show standard deviations. (C) The fraction of cells with distributions consistent with dropout across layers and transgenic mouse lines in primary visual cortex (VISp). Error bars show standard deviations.

used during training, and turned off during evaluation. Here, we continued to use dropout during evaluation, which introduces variability in the responses of each unit. Using dropout in this way has been proposed as a form of Bayesian approximation [10]. Critically, we find that the bimodal distribution in neural responses across trials can also be captured by a convolutional neural network trained with dropout (compare Fig 5b and Fig 5c).

**Control with minimum reliability threshold.** We also quantified neural variability at the population level. One measure of response reliability is the mean trial-to-trial correlation of neural activity within a session, which is bounded between 0 (low reliability) and 1 (high reliability) [8]. Over the entire population of cells, we find very low mean response reliabilities across sessions (Figure 6a), with a mean reliability of 0.11. To remove extremely unreliable cells from our analysis, we set a reliability threshold of 0.1. Our subsequent analyses are performed using this set of cells ($N = 1775$), which has a mean reliability of 0.37.

**Dropout-like response distributions.** To identify cells with dropout-like response distributions, we first found the cells that were better fit by Gaussian mixture models with two components, and for each of these cells, we computed a z-score on the component with the lower mean to test whether it was significantly different from zero. For cells which were deemed better fit by the two-component Gaussian mixture model, we performed an additional test to determine whether their response distributions were dropout-like. For each of these cells, we calculated a z-score on the component with the lower mean, and those cells with z-scores less than two (meaning their means are

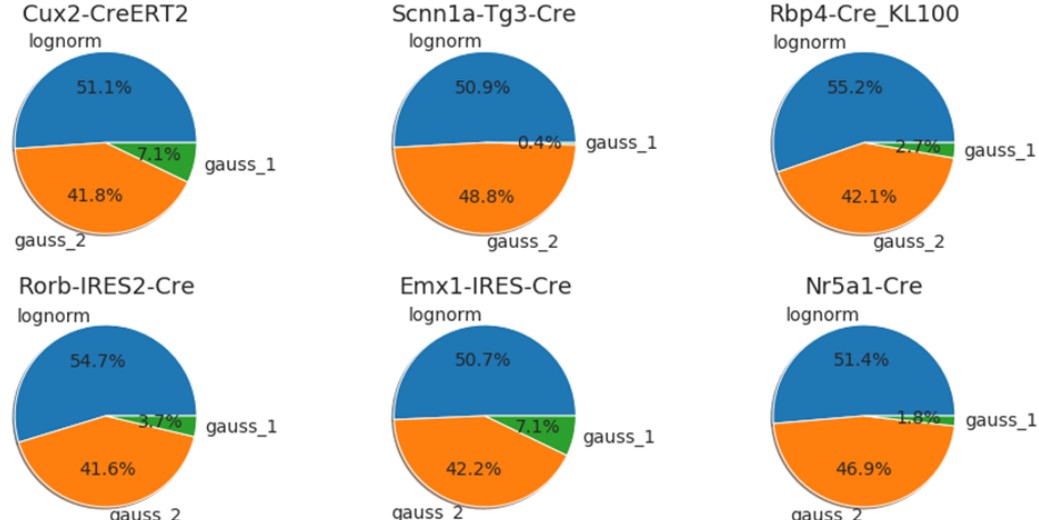

Figure 7: Dependence of noise distribution on cell types. We found relatively consistent results across all Cre-lines tested. The selected Cre-lines selectively label excitatory cells in different areas and layers.

not significantly different than zero) were counted as cells with dropout-like response distributions. The number of cells that pass these two tests divided by the total number of cells gives the fraction of cells which have dropout-like response distributions.

**Additional analyses.** We also performed additional analyses, separating cells by visual area, layer, and transgenic mouse line. Based on this measure, we find that the fraction of cells with dropout-like response distributions is high and relatively constant across visual areas, with a mean fraction of 0.74 (Figure 6b). We also find only modest differences in the fraction of cells with dropout-like distributions across layers and mouse transgenic lines, with the lowest fraction of cells being in the superficial layers (Figure 6c). Finally, we also studied the noise distributions across Cre-lines in our dataset (Figure 7). We found consistent results across all Cre-lines, which selectively labeled excitatory cells in different cortical areas and layers.

# mouse 1

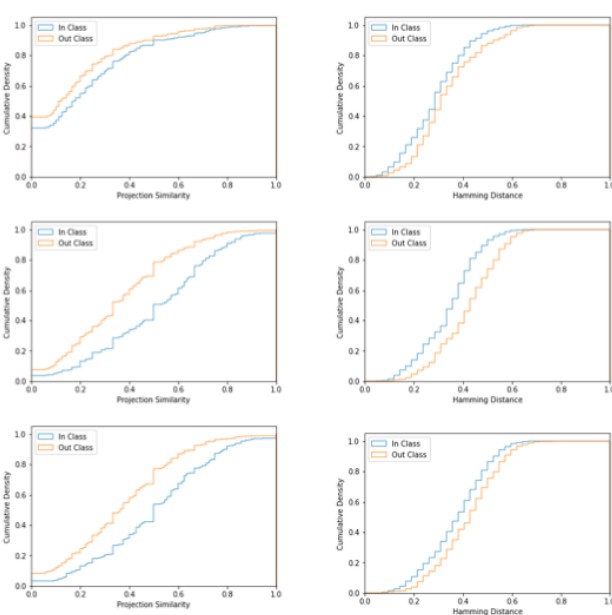

Figure 8: Cumulative distributions for similarity metrics using data aggregated across areas for mouse 1. Projection similarity (left column) and Hamming distance (right column) between clip variance subspace $V_k$ and noise subspace $N_{j,k,n}$ (line 1), between clip coding subspace $C_k$ and noise subspace $N_{j,k,n}$ (line 2) and between exemplar coding subspace $E_k$ and noise subspace $N_{j,k,n}$ (line 3). All the observed differences are statistically significant (KS test, $p < 0.05$).

# mouse 2

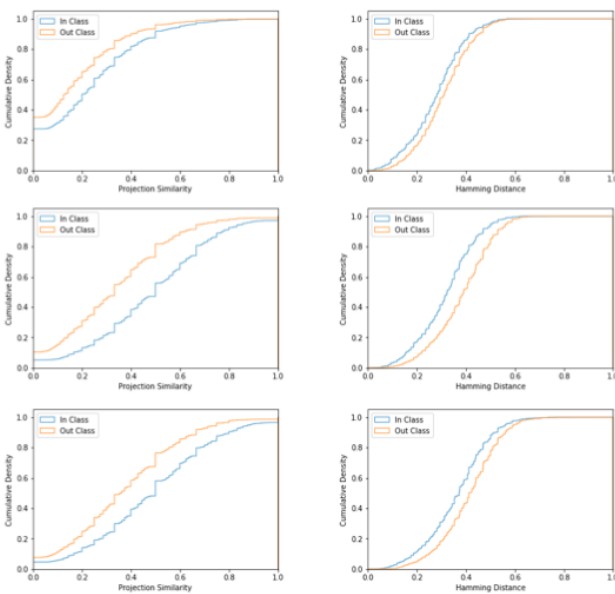

Figure 9: Cumulative distributions for similarity metrics using data aggregated across areas for mouse 2. Projection similarity (left column) and Hamming distance (right column) between clip variance subspace $V_k$ and noise subspace $N_{j,k,n}$ (line 1), between clip coding subspace $C_k$ and noise subspace $N_{j,k,n}$ (line 2) and between exemplar coding subspace $E_k$ and noise subspace $N_{j,k,n}$ (line 3). All the observed differences are statistically significant (KS test, $p < 0.05$).

# mouse 3

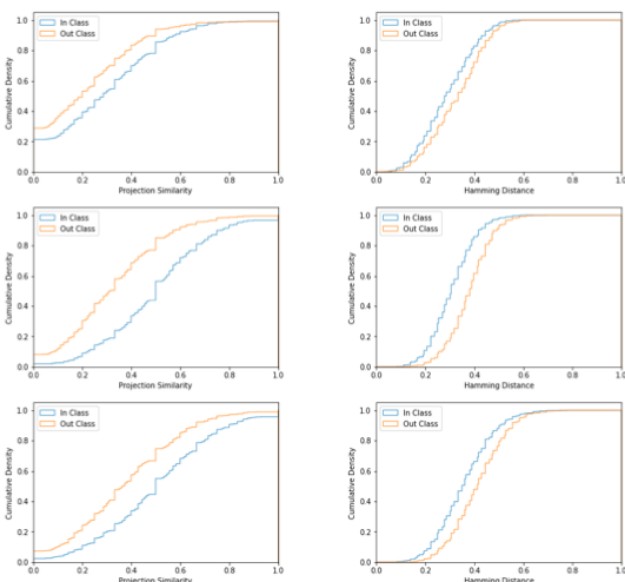

Figure 10: Cumulative distributions for similarity metrics using data aggregated across areas for mouse 3. Projection similarity (left column) and Hamming distance (right column) between clip variance subspace $V_k$ and noise subspace $N_{j,k,n}$ (line 1), between clip coding subspace $C_k$ and noise subspace $N_{j,k,n}$ (line 2) and between exemplar coding subspace $E_k$ and noise subspace $N_{j,k,n}$ (line 3). All the observed differences are statistically significant (KS test, $p < 0.05$).

