# OpenReview forum: "Does the neuronal noise in cortex help generalization?"
_NeurIPS.cc/2019/Workshop/Neuro_AI — Real Neurons & Hidden Units @ NeurIPS 2019 Poster_

### Official Review · AnonReviewer2 · 2019-09-24
**Dropout inspires a hypothesis on the functional role for noise in generalization, but analysis is superficial.**

**Clarity:** 2

**Comment:**

AI->Neuro:
"We believe that research into the structure and role of biological noise will be useful for developing new methods to train neural networks with better generalization capabilities."
The inspiration appears to have propagated in the opposite direction, i.e. CNNs with dropout are proposed as a null model which seems to explain the observations in the data.

"Future work will study how different forms of subspace-aligned noise may help deep neural networks generalize better from fewer examples."
This is an interesting prospect of the work proposed in the abstract, but essentially left for future investigation.

With this in mind, while the analysis of the noise properties of neural reponses in these datasets is important for the field of Neuroscience, descriptions of the datasets used and control for eye movements seems to be lacking.  This work appears to offer little new insight for the AI field.


**Category:**

AI->Neuro

**Clarity Comment:**

See rigor.

**Evaluation:**

2: Poor

**Importance:**

2: Marginally important

**Importance Comment:**

CNNs with dropout are proposed as a null model which seems to explain the response noise in neural data, but CNNs have no eye movements, while the neural data apparently does.

**Intersection:**

3: Medium

**Intersection Comment:**

While the analysis of the noise properties of neural responses in these datasets is important for the field of Neuroscience, descriptions of the datasets used and control for eye movements seems to be lacking.  This work appears to offer little new insight for the AI field in its current preliminary form.

**Rigor Comment:**

deltaF/F - explain.
For neuroAI symposium, better to define these terms.

Details on convnet too sparse.  Maxpooling layers? What was achieved performance on CIFAR-10 test set?

I would liked to have seen a log-norm example class in Fig 1.
The mixture of Gaussians depicted in Fig 1 (a), is one to consider <0.1 signal, or noise?  Perhaps its no response plus noise? Do you have any baseline "no stimulus" epochs to quantify the level of baseline noise for each of the units considered?

For Allen Institute data, and the head fixed mice, apparently eye movements were not paralyzed.
This is an important source of variability which is not accounted for in the CNN model.
i.e. for a given presentation of the stimulus, one has no idea if receptive fields are receiving remotely similar illumination.
Perhaps a way to proceed to control for this would be to inject the Allen Institute recorded eye movements into the CNN model input stream.



**Technical Rigor:**

2: Marginally convincing

---

### Official Review · AnonReviewer3 · 2019-09-24
**Tries to link observations of noise in cortex to dropout, but relationship is unclear**

**Clarity:** 2

**Comment:**

There are a number of claims that don't appear to be very well supported by what is actually shown.

**Category:**

AI->Neuro

**Clarity Comment:**

Quite difficult to follow the chain of reasoning here.

**Evaluation:**

1: Very poor

**Importance:**

1: Irrelevant

**Importance Comment:**

Tries to argue based on which distributions fit best the noise measured in cortex that the networks are similar to ANNs with dropout. However, the link is tenuous and not strongly argued.

**Intersection:**

2: Low

**Intersection Comment:**

In principle showing that dropout was like neuronal noise could be interesting, but it's pretty tenuous here.

**Rigor Comment:**

It's a weak link to show that two distributions are similar, especially when only a very few distributions were fit and there are a lot of missing details about what is meant by a "dropout-like" distribution (early on it says "see Methods" but then isn't defined in the Methods).

**Technical Rigor:**

1: Not convincing

---

### Official Review · AnonReviewer1 · 2019-09-26
**Quantification of neural noise is poorly described, and alternative explanations not explored**

**Clarity:** 3

**Category:**

AI->Neuro

**Clarity Comment:**

Figure 3 needs much more description, are the subspaces defined using trial-averaged responses? Why define a noise subspace instead of looking at the direction of the noise on each trial?

**Evaluation:**

2: Poor

**Importance:**

1: Irrelevant

**Importance Comment:**

The authors suggest that noise corresponds to a regularization step that the brain does, like dropout. Single neuron noise could result in information from that neuron not propagating further down the network, like in dropout (but not equivalent for various reasons), but that would require more thoughtful comparisons with the neural activity, instead of just a comparison of distribution of activations.

**Intersection:**

2: Low

**Intersection Comment:**

Comparisons are too preliminary.

**Rigor Comment:**

Controlling for state variables was a useful step to perform. Otherwise the comparisons are not well-quantified and other explanations of the noise/models are not explored.

**Technical Rigor:**

2: Marginally convincing

---

### Decision · Program_Chairs · 2019-10-02

Accept (Poster)